# Motivation and physical activity across Chinese adolescents: Based on latent profile analysis

**Ying Zhao[1], Qinghua Wu[2], Wei Zheng[1]***

**1** School of Physical Education and Health, Sanming University, Sanming, Fujian, China, **2** Graduate School, Guangzhou Sport University, Guangzhou, Guangdong, China

* zhengweiSanmingUniversity@outlook.com

## Abstract

Despite known links between motivation and physical activity, latent profiles of motivation among Chinese adolescents remain unexamined. Using the person-centered approach, this paper explores the relationship between adolescent motivation and physical activity. We aim to identify the latent motivation profiles and examine how these profiles differential predict physical activity (PA) levels, with attention to gender and age variations. This study recruited 571 adolescents ($M_{age}$ = 11.995, $SD$ = 1.519) in southern China by the scale of Sport Motivation Scale and International Physical Activity Questionnaire-Short Form. We conducted latent profile analysis (LPA) to classify motivation subgroups using Mplus. MANOVA and ANOVA were employed to compare PA differences across profiles, genders, and education levels. The results indicate that three profile model is the optimal model: Low Motivation-High Amotivation (8.45%), Moderate Motivation-High Amotivation (60.61%), and High Autonomous Motivation (30.94%). The subgroup with higher scores of intrinsic motivation and external motivation reported more PA. Moreover, male's PA is significantly more active than female adolescents, while older adolescents have less PA than younger adolescents. This study identified adolescents with different motivation profiles and PA. Findings suggest the need for more personalized strategies to promote adolescent participation in PA and provide a novel insight into intervention for adolescents with low motivation. Further research could be measured by objective methods and long-term design.

## Introduction

Moderate physical activity (PA) has a positive impact on the health of adolescents, a recommendation from WHO that children and adolescents should spend 60 minutes on moderate-vigorous physical activity (MVPA) per day [1]. However, about 80% of adolescents lack daily PA [2] and Insufficient PA may lead to diseases and obesity. This issue has also attracted widespread attention in China [3,4].

**Data availability statement:** All necessary to replicate this study's findings are publicly available without restriction in the Figshare repository: https://doi.org/10.6084/m9.figshare.29316266.v1.

**Funding:** This work was supported by the Scientific Research Start-up Fund for High-level Talents of Sanming University (Grant No. 25YG12S), awarded to Assistant Professor Ying Zhao.

**Competing interests:** No authors have competing interests.

Therefore, the purpose of this study is to explore the relationship between motivation and PA among adolescents. In particular, the present study used the person-centered approach to explore the latent profiles of motivation and examine the differences in PA among each subgroup. Results can provide a novel insight into intervention for PA among adolescents.

## Motivation and physical activity

PA can positively influence people's physical health [5,6]. Especially, PA is especially beneficial for weight control, the development of the cardiovascular system, and cancer prevention [7]. In addition, PA also has positive effects on people's mental health, such as promoting psychological health and cognitive abilities [8]. Therefore, WHO and many countries are recommending adolescents engage in PA and maintain this habit [9], which is beneficial for their health and academic achievement.

Unfortunately, adolescents are relatively inactive in many countries [10]. Early studies found that lacking PA is a growing problem among children and adolescents in China [11], which has a negative impact on the health of teenagers [12]. Although the Chinese government has tried various ways to increase PA among students [13], there are still many students who are in a state of insufficient physical activity. In particular, male students preferred PA to female students [9,14] and younger students' engagement in sports was more frequent than older students [15]. This phenomenon was underpinned by various key variables, which needed to be identified.

Relative research suggests that both external and internal factors can affect adolescents' engagement with PA. External factors, such as parents and peers, are exactly vital for students' PA. For example, parental encouragement and support positively relate to children and adolescent PA [16]. Their parents mainly initiate children's sports participation at the beginning and parental support is necessary for children and adolescents during participation [17,18]. Although parental influence is important to children, this influence will shift to classmates when children get older [19], supporting that the influence from peers can affect young adolescents' PA engagement effectively [20]. For example, peer support can promote adolescent participation in sports [21].

Internally, based on the Self-Determination Theory (SDT), previous studies have shown that motivation is vital for PA [22]. Motivation can be divided into intrinsic motivation, external motivation, and amotivation [23]. In particular, intrinsic motivation is defined as the engagement of activity for its inherent satisfaction [24], which can be divided into three types. Firstly, intrinsic motivation to know relates to engagement for the satisfaction of learning, exploration, and understanding. Secondly, intrinsic motivation toward accomplishments is defined as engagement for unique accomplishments. Finally, intrinsic motivation to experience stimulation is people's engagement in their stimulating feelings [25].

Additionally, external motivation refers to doing something because of external value or regulation and external motivation can be divided into three types. Initially, external regulation is that people's behavior is controlled by external resources. Furthermore, with introjection, people to some extent agree with external perspectives,

and their behavior is not needed to initiate anymore. In addition, in the next stage, identification refers to people performing with internal regulation and self-determination, although they are still motivated by extrinsic reasons [25]. Unlike intrinsic and external motivation, amotivation is when people believe something has no value so they either do not act at all or act without intention [23].

Motivation is a key factor in PA. Early studies identified a positive relationship between motivation and adolescents' PA [26,27], adolescents with higher motivation for exercise are more likely to maintain good exercise habits. For example [28], found that adolescents' motivations are positively associated with MVPA, and this relationship is moderated by settings. Additionally [14], has shown the positive effects of intrinsic motivation on PA among Chinese students, which emphasized the importance of intrinsic motivation.

However, motivation can positively promote PA but relatively research examines the influence of different types of motivations at the same time. Moreover, although many Chinese scholars focused on the relationship between motivation and PA based on the variable-centered, relatively less scholars explored the latent profile of motivation among adolescents by a person-centered approach, which can offer a novel sight for intervention in physical activity.

## Variable-centered approach and person-centered approach

The variable-centered approach is popular in social science, which aims to explore the relationship between variables. This method assumes that people in the same sample are a set of "averaged" parameters [29]. However, the person-centered approach provides another perspective to examine the assumption of people homogeneity [30].

The person-centered approach assumes that the sample could be divided into several subpopulations by different sets of parameters [29], which is becoming popular. This approach aims to identify the latent subgroup with a common feature based on a specific subset of variables [30]. The strength of this approach is that it can provide finer-grained detail about the sample [31] therefore it is suitable for research that aims to categorize latent subgroups and explore the differences of subgroups.

The person-centered approach can be divided into latent class analysis (LCA), latent profiles analysis (LPA), and so on. LCA is used for discrete observed variables, whereas LPA is used for continuous observed variables. LPA has received lots of attention and interest recently [32] and is practical in the area of physical activity [33,34]. For example, a study based on Spanish handball players identified distinctive profiles of motivation, fear, and anxiety and examined the differences between the profiles by covariables and specific playing positions [35]. In addition to athletes, some scholars also used latent profile analysis to explore students' motivation for physical activity. Haerens's study investigated the relationship between college students' physical motivation by latent profile analysis and results found that students with more optimal motivational profiles reported more engagement in PA [36]. Another study applied latent profile analysis to identify distinct motivational profiles for PA among adults [37].

While prior studies have extensively examined the relationship between motivation and PA using variable-centered approaches [22,38], few have adopted a person-centered perspective to identify heterogeneous motivation profiles among Chinese adolescents. The unique academic pressure and sociocultural context in China may influence the distribution of adolescents' sports motivation and enthusiasm [39]. Existing research in China predominantly focuses on linear associations [14], neglecting potential subgroups with distinct motivational patterns. This study addresses these gaps by employing LPA to classify motivation subtypes and linking them to physical activity behaviors, thereby offering targeted intervention insights.

## The current research

This study examined the latent profiles of motivation among Chinese students and explored the difference in subgroups in physical activity by LPA. Therefore, this study assumes that (a) different categories of motivation are present in students and (b) there may be a significant difference in PA in different latent categories.

## Methods

### Participants

This research was approved by the institutional ethical committee of the university (IRB: 2025LCLL-009), students, school, their parents or guardians. Under the guidance of the teacher, students are required to complete the questionnaire within 20 minutes between 18th January, 2025 and 20th January, 2025. A total of 641 students were recruited from one primary school (5th and 6th) and one junior high school (7th) in the south of China and all students received a small gift after they finished the survey. We did not include a questionnaire that was blank or gave the same response [40] and the final sample was 571 ($M_{age}$ = 11.995, $SD$ = 1.519, Age range = 10–14, Male = 279, Female = 296, Primary school student = 344, Junior high school student = 277). Based on past research, samples of 300–500 would qualify in latent profile analysis [41,42], and samples of 500 can lead to enough accuracy [32], this research satisfied the requirements of sample size.

### Measures

**Motivation.** Motivation was measured by the Chinese version of the Sports Motivation Scale, which was modified from the Sport Motivation Scale [25,43]. The Chinese version of the Sports Motivation Scale consists of 7 dimensions: intrinsic motivation to know (4 items, such as for the pleasure it gives me to know more about the sport that I practice, α = 0.718), intrinsic motivation to accomplishment (4 items, α = 0.764), intrinsic motivation to experience stimulation (4 items, α = 0.762), introjection (4 items, α = 0.707), identification (4 items, α = 0.708), external regulation (4 items, α = 0.741) and amotivation (4 items, α = 0.720). The scale used a 5-point Likert scale, where 1 indicates "Not agree at all" and 5 indicates "Very strongly agree". This scale has demonstrated good applicability in Chinese research [44].

**Physical activity.** Physical activity was assessed metabolic equivalent of task (MET) of the student and measured by the International Physical Activity Questionnaire short form (IPAQ-SF), which consists of 7 items [45]. Six items investigate the participants' physical activity, and one item examines the duration of sedentary behavior. The weekly MET for each participant is calculated as the sum of the activity type value × weekly frequency × daily duration. Especially, MET in this research was calculated as 8.8 (vigorous-intensity activity) × weekly frequency × daily duration + 4.0 (moderate-intensity activity) × weekly frequency × daily duration + 3.3 (moderate-intensity) × weekly frequency × daily duration [46,47]. IPAQ-SF is considered effective and recommended in many countries [48,49]. In this study, the calculated MET values are standardized before further analysis, and the extreme values (less than −3.29 or more than +3.29) will be eliminated [50].

**Demographic variables.** Demographic variables included students' gender (male = 1, female = 2), age, and education stage (primary school student = 1, junior high school student = 2).

### Statistical analysis

Descriptive statistics, correlations, analysis of variance (ANOVA), and multivariate analysis of variance (MANOVA) were analyzed by SPSS 25.0. Moreover, this research used Mplus 8.3 to conduct an LPA of the scores on the seven dimensions of sports motivation.

This research examined 1−6 profiles to explore the optimal profile based on the recommendation of model fit [41]. The optimal profile model is determined by AIC, BIC, aBIC, LMR, BLRT, and Entropy [32]. The smaller values of AIC, BIC, and aBIC indicate the better model fit. The optimal profile model was determined by comparing model fit indices including AIC, BIC, and aBIC. Following established guidelines, smaller absolute values of these indices indicate better model fit. Specifically: AIC values typically range from thousands to tens of thousands in large samples, with differences >10 suggesting substantial improvement. BIC and aBIC are more stringent, with differences >6 considered meaningful [32,42]. Additionally, if the values of BLRT and LMR are significant, the model fit of the $k$ category is significantly better than the $k-1$ category. Finally, entropy represents the accuracy of the model category and the values of 0.8 or greater suggest the high accuracy of the model category [41,42].

## Results

### Multicollinearity and common methods bias

Multicollinearity represents a high degree of linear intercorrelation between explanatory variables and leads to incorrect results of regression analyses [51] and this research used variance inflation factor (VIF) to examine the multicollinearity. The results of VIF were 1.156–3.411 (Table 1), which showed that there was no severe multicollinearity in this research.

Common method bias (CMB) is a potential problem in behavior research [52] and may cause a negative impact on the relationships between constructs [53]. This research used Harman's single-factor test to measure the CMB and the result showed account for the variance in the variables is 35.351%, which illustrated that this research did not have a substantial amount of common method variance.

### Descriptive statistics and correlation analysis

The results of descriptive statistics and Pearson's correlation analysis are shown in Table 2. Intrinsic motivation to know was positively correlated with intrinsic motivation to accomplishment ($r = .667$, $p < .001$), intrinsic motivation to experience stimulation ($r = .737$, $p < .001$), introjection ($r = .662$, $p < .001$), identification ($r = .585$, $p < .001$), external regulation ($r = .542$, $p < .001$), and physical activity ($r = .156$, $p < .001$). Intrinsic motivation to accomplishment was positively correlated with intrinsic motivation to experience stimulation ($r = .757$, $p < .001$), introjection ($r = .725$, $p < .001$), identification ($r = .643$, $p < .001$), external regulation ($r = .648$, $p < .001$), and physical activity ($r = .152$, $p < .001$). Intrinsic motivation to experience

**Table 1. Results of VIF.**

| | Variables | Tolerance | VIF |
|---|---|---|---|
| 1 | Intrinsic Motivation to Know | 0.405 | 2.469 |
| 2 | Intrinsic Motivation to Accomplishment | 0.322 | 3.104 |
| 3 | Intrinsic Motivation to Experience Stimulation | 0.294 | 3.411 |
| 4 | Introjected Regulation | 0.360 | 2.779 |
| 5 | Identified Regulation | 0.424 | 2.361 |
| 6 | External Regulation | 0.417 | 2.396 |
| 7 | Amotivation | 0.865 | 1.156 |

**Table 2. Descriptive statistics and correlations of key variables.**

| | Variables | *M* | *SD* | Ske. | Kur. | 1 | 2 | 3 | 4 | 5 | 6 | 7 |
|---|---|---|---|---|---|---|---|---|---|---|---|---|
| 1 | Intrinsic Motivation to Know | 3.627 | 0.774 | −0.738 | 1.145 | | | | | | | |
| 2 | Intrinsic Motivation to Accomplishment | 3.656 | 0.826 | −0.816 | 1.135 | .667*** | | | | | | |
| 3 | Intrinsic Motivation to Experience Stimulation | 3.461 | 0.849 | −0.502 | 0.399 | .737*** | .757*** | | | | | |
| 4 | Introjection | 3.575 | 0.785 | −0.577 | 0.906 | .662*** | .725*** | .715*** | | | | |
| 5 | Identification | 3.443 | 0.751 | −0.647 | 1.176 | .585*** | .643*** | .636** | .626*** | | | |
| 6 | External Regulation | 3.432 | 0.825 | −0.597 | 0.790 | .542*** | .648*** | .660** | .666*** | .668*** | | |
| 7 | Amotivation | 2.917 | 0.882 | −0.103 | −0.344 | −.022 | .038 | .022 | .081 | .244*** | .226*** | |
| 8 | Physical activity | −0.064 | 0.808 | 1.687 | 2.977 | .156** | .152** | .172*** | .159*** | .106* | .133** | −.063 |

Notes. *p < 0.05, **p < 0.01, ***p < 0.001. Ske. = Skewness, Kur. = Kurtosis.

stimulation was positively correlated with introjection ($r=.715$, $p<.001$), identification ($r=.636$, $p<.001$), external regulation ($r=.660$, $p<.001$), and physical activity ($r=.172$, $p<.001$).

Introjection was positively correlated with identification ($r=.626$, $p<.001$), external regulation ($r=.666$, $p<.001$) and physical activity ($r=.159$, $p<.001$). Identification was positively correlated with external regulation ($r=.668$, $p<.001$), amotivation ($r=.224$, $p<.001$), and physical activity ($r=.106$, $p=.012$). External regulation was positively correlated with amotivation ($r=.226$, $p<.001$), and physical activity ($r=.133$, $p<.001$).

However, amotivation showed no significant correlations with intrinsic motivation to accomplishment ($r=-.022$, $p=.610$), intrinsic motivation to experience stimulation ($r=.038$, $p=.365$), introjection ($r=.022$, $p=.606$), identification ($r=.081$, $p=.056$), and physical activity ($r=-.063$, $p=.134$).

## Latent profile analysis of motivation

This research used LPA to divide motivation of PA into categories and the model fitting indices of 1–6 categories are shown in Table 3. The values of AIC, BIC, and aBIC were shapely decrease from 1 category to 3 categories whereas slowly decreasing from 4 categories to 6 categories, indicating that the model with more categories was better. However, the $p$-value of the LMR of the 4-category model is more than.05 which indicating that the model with 3-category model was no significantly better than 3-category model. Finally, the value of entropy in 3-category model is more than 0.8, indicating the high accuracy of latent profile. Overall, category 3 could be considered as the optimal latent profile model according to the above parameters.

Additionally, the distribution of scores on the seven dimensions of the 3-category model is shown in Fig 1. Firstly, Profile 1 had the smallest number of students (8.45%), 48 students, and the lowest scores in intrinsic motivation and external motivation but their score of amotivation was relatively high than other dimensions. This result showed that this group of students lacks intrinsic motivation and external motivation and they do not feel competent in physical activity. Therefore, this group was named as "Low Motivation-High Amotivation". Secondly, Profile 2, with a total of 345 students (60.61%), had moderate scores in intrinsic motivation and external motivation but their score of amotivation was the highest, so it was named "Moderate Motivation-High Amotivation". Finally, Profile 3 (30.94% and 178 students) scored significantly higher than other two group in intrinsic motivation and external motivation whereas their score in amotivation was lower than Profile 3, which was named "High Autonomous Motivation".

Moreover, this research conducted a covariate analysis (gender and education stage) in the 3-category model (Table 4). Results suggested gender was not significantly different between Profile 1 (Low Motivation-High Amotivation) and Profile 2 (Moderate Motivation-High Amotivation) whereas significantly different between Profile 1 and Profile 3 (High Autonomous Motivation), with more male students represented in the high motivation group. The coefficient of Profile 1 was bigger than Profile 3, indicating that the score of male motivation in Profile 3 was higher than Profile 1. For education, scores were also not statistically significantly different between Profile 1 and Profile 2 but significantly different for Profile 1 as compared to Profile 3. Negative coefficients indicated the scores of primary school students in the high-motivation group were higher than the low-motivation group.

**Table 3. Latent profile model information criteria, likelihood ratio test, and entropy ($n=571$).**

| Model | K | Log (L) | AIC | BIC | aBIC | Entropy | BLRT | LMR | Group size |
|---|---|---|---|---|---|---|---|---|---|
| 1 | 14 | −4765.965 | 9559.930 | 9620.794 | 9576.350 | / | / | / | 1 |
| 2 | 22 | −4199.404 | 8442.807 | 8538.450 | 8468.610 | .905 | <.001 | .0334 | 0.1821/0.8179 |
| 3 | 30 | −3831.872 | 7723.744 | 7854.165 | 7758.929 | .886 | <.001 | .0019 | 0.0845/0.6061/0.3094 |
| 4 | 38 | −3676.141 | 7428.282 | 7593.483 | 7472.850 | .860 | <.001 | .2273 | 0.0583/0.5131/0.2795/0.1492 |
| 5 | 46 | −3591.447 | 7274.894 | 7474.874 | 7328.844 | .852 | <.001 | .1287 | 0.0366/0.0989/0.3609/0.3847/0.1189 |
| 6 | 54 | −3533.188 | 7174.377 | 7409.136 | 7237.710 | .830 | <.001 | .5602 | 0.0337/0.0871/0.3321/0.3043/0.1347/0.1081 |

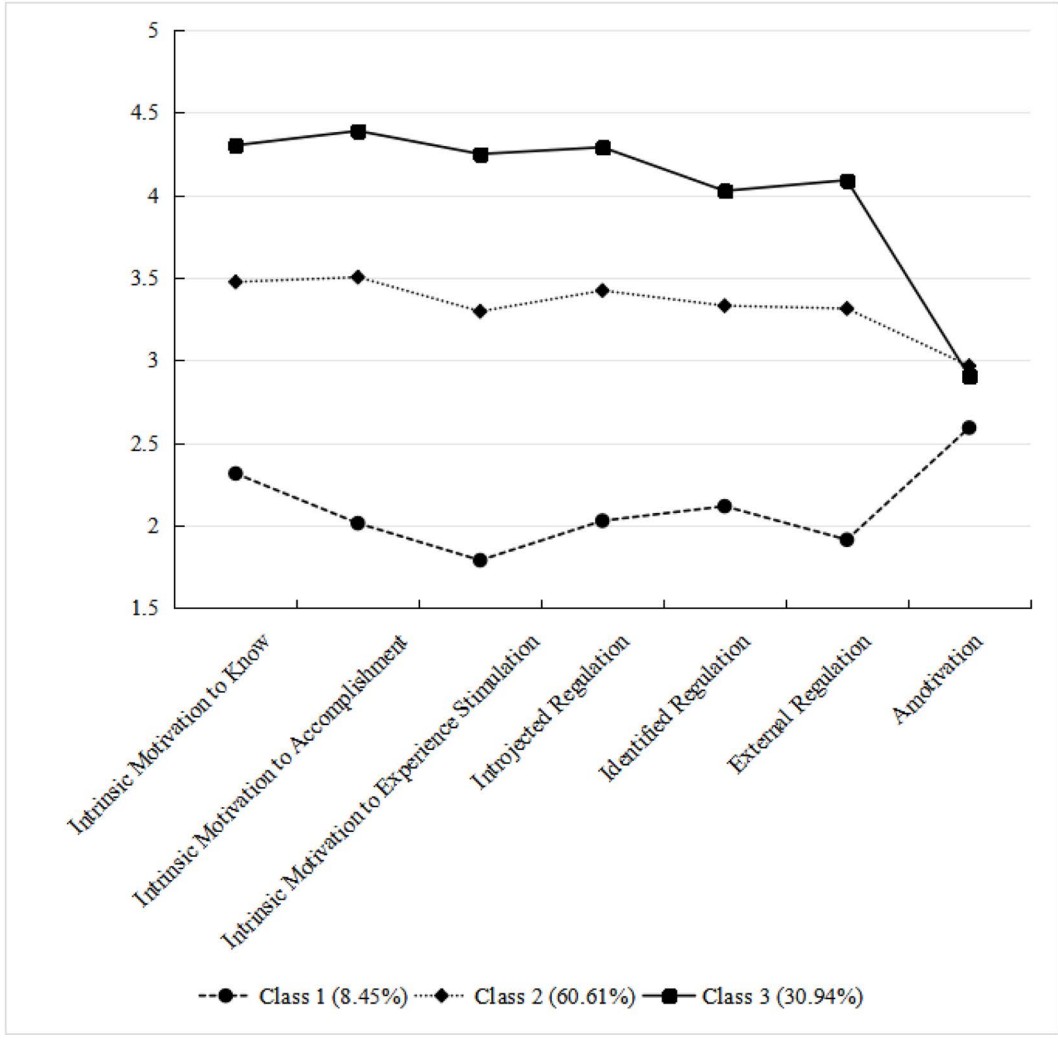

**Fig 1. Motivational profiles of the best fitting model (4 Profiles).**

## Relationship between covariate variables and physical activity

MANOVA was used to compare the differences in gender and education in PA (Table 5). When gender is used as an independent variable, the scores of males' PA are significantly more than females ($F$ (3, 364) = 4.714, $p$ = .030). When age is used as an independent variable, there is no significant difference between primary school students and junior high school students ($F$ (3, 364) = 1.799, $p$ = .180), although the scores of primary school students ($M$ = −0.0001, $SD$ = 0.8253) were higher than junior high school students ($M$ = −0.1570, $SD$ = 0.7740).

## Relationship between motivational profiles and physical activity

One-way ANOVA was used to compare the differences between 3 profiles in PA (Table 6) and results showed that differences across the 3 profiles ($F$ (2, 565) = 12.523, $p$ < .001). Profile 1, with the lowest score in physical activity, was significantly lower than Profile 3 ($p$ < .001) whereas not significantly lower than Profile 2 ($p$ = .405). Moreover, Profile 2 was significantly lower than Profile 3 ($p$ < .001).

**Table 4. Covariate analysis results for four-profile model.**

| Variable | Group 2<br>Moderate Motivation-High Amotivation<br>(*n*=345) | Group 3<br>High Autonomous<br>Motivation<br>(*n*=136) |
|---|---|---|
| Gender | −0.134 | −0.957** |
| Education Stage | −0.205 | −1.363** |

*Note.* Group 1 (Low Motivation-High Amotivation, *n*=48) served as the reference group. ** *p*<0.01.

**Table 5. Results of MANOVA of gender and education.**

| Variable | *M (SD)* | | *F* | *p* |
|---|---|---|---|---|
| | **Male** | **Female** | | |
| Physical Activity | −0.0146 (0.8460) | −0.1077 (0.7700) | 4.714 | .030 |
| | **Primary School Students** | **Middle School Students** | | |
| Physical Activity | −0.0001 (0.8253) | −0.1570 (0.7740) | 1.799 | .180 |

# Discussion

Our findings extend previous variable-centered research by revealing three distinct motivation profiles among Chinese adolescents, a classification not previously documented in this population. The identification of the 'Low Motivation-High Amotivation' subgroup (8.45%) highlights a critical target for interventions, which earlier studies might overlook when treating motivation as a homogeneous construct.

## Latent categories of motivation in physical activity

The results indicated that there were three subgroups of motivation in PA, namely, Profile 1 (Low Motivation-High Amotivation), Profile 2 (Moderate Motivation-High Amotivation), and Profile 3 (High Autonomous Motivation).

Profile 1 accounts for 8.45%, whose scores in motivation are much lower than any other profile. In particular, the score of intrinsic motivation to experience stimulation in this subgroup is lower than any other dimensions and the score of amotivation is the highest, which means that adolescents in this subgroup lack motivation or intention for PA. Although this group has the smallest number of adolescents, it still requires the attention of parents and teachers and needs to improve sports motivation.

Profile 2 has the highest number of students (60.61%) and its scores are more even in each dimension except for amotivation and the score of intrinsic motivation to accomplishment is the highest. The result indicates that the intrinsic motivation and external motivation of adolescents in this subgroup are developed in a balanced way.

**Table 6. Results of ANOVA of LPA.**

| Variables | *n* | *M* | *SD* | Grade | *p* |
|---|---|---|---|---|---|
| Group 1<br>(Low Motivation-High Amotivation) | 48 | −0.3091 | 0.7073 | Group 2 | .405 |
| | | | | Group 3 | <.001 |
| Group 2<br>(Moderate Motivation-High Amotivation) | 345 | −0.1526 | 0.7471 | Group 3 | <.001 |
| Group 3<br>(High Autonomous Motivation) | 178 | 0.1759 | 0.8926 | / | / |

Additionally, the students of Profile 3, which is 30.94%, are less than Profile 2 and have the highest scores in intrinsic motivation and external motivation. The score of intrinsic motivation to accomplishment is the highest and the score of amotivation is much lower than other dimensions in this subgroup, which means that adolescents in this subgroup prefer to engage in PA.

Based on covariate analysis, there are no significant between Profile 1 and Profile 2 whereas there are significant differences between Profile 1 and Profile 3 both in gender and education. These results indicate that males are more active than females and primary school students are more active than junior high school students in motivation, consistent with previous findings. For example, a study based on Portuguese junior high school students showed that boys show higher motivation than girls and older students show lower motivation [54]. Another Chilean study found that male adolescents had statistically significantly more self-determined motivation toward exercise than female adolescents [55].

There may be several reasons for gender differences in motivation. Firstly, gender stereotypes can influence teenagers' choices and participation in sports. During childhood, children may develop stereotypes about sports influenced by their environment [56,57], such as those reinforced by social media or television viewing, which can lead to beliefs about gender limitations in certain domains [58]. Moreover, the physiological differences between boys and girls can also explain this difference to a certain extent [59], boys' growth rate begins to accelerate during adolescence, gradually becoming stronger than girls in abilities such as strength and speed. This may make girls realize that boys may be more suitable for sports and weaken their motivation for physical activity.

## Difference of gender and education in physical activity

Early studies used the variable-centered approach to explore the relationship between motivation and other variables about sports. Relatively fewer scholars in China focus on motivation and PA among adolescents from a person-centered perspective, which could provide some specific information.

The results of gender and education stage differences in PA were consistent with previous research, which has shown that male adolescents were more active than female adolescents and the PA of junior high school students was lower than that of elementary school students [15,60,61].

According to developmental theory, the reason why girls are less active than boys can be divided into peers and parents [20]. Girls are more likely to experience teasing from their peers during PA [61], which may negatively influence their flow and enjoyment [62] and decrease girls' sports engagement. Moreover, in addition to peers, their parents influence teenagers' behavior [63]. Parents have different expectations for boys and girls, which can affect their support for their children's PA [64]. Previous studies in China have found that girls received less parental support, which had a negative impact on their exercise behavior [60].

The reason for the phenomenon that adolescents' PA declines substantially with their age may be academic pressure [39]. Junior high school students are facing greater academic pressure so they may have less time for PA. Another potential reason is parental influence. The influence of parents is important in children and adolescents' participation in sports [60]. An early study found that younger children presented higher parental support from their parents and reported more physical activity than older children [65].

## Relationship between optimistic latent profiles and physical activity

In addition, the ANOVA results indicated that the subgroup with higher scores participated in PA more frequently, which was consistent with the SDT and previous research, which have shown a positive relationship between self-determined motivation and PA among adolescents.

The results of latent profile analysis and ANOVA have shown that compared with other profiles, the motivation and PA of Profile 1 (Low Motivation-High Amotivation) were the lowest. This disparity may lead to "health inequality" phenomena and affect students' health negatively in this subgroup, although this subgroup had the fewest students. In particular,

the score of amotivation was relatively higher than any other dimension in this subgroup. Additionally, the score of intrinsic motivation and external motivation was lower than Profile 2 and Profile 3. Amotivation is the result of people feeling that doing something is meaningless or not valuable [66], and people with amotivation may not have the willingness to exercise [24]. The previous study pointed out that amotivation had been correlated with dropout among athletes and low engagement of students in PA [67]. Another recent study found that amotivation was associated with lower self-reported MVPA [68]. Therefore, it is necessary to decrease amotivation by raising their awareness of PA and increasing their ability beliefs [67].

Additionally, the PA of Profile 2 (Moderate Motivation-High Amotivation) ranked second among the three subgroups and had the largest number of students. The score of amotivation was the highest compared with others' profiles, which could negatively impact a student's PA. However, the scores of intrinsic motivation and external motivation were relatively higher than amotivation and intrinsic motivation was higher than external motivation. A previous study found that students' intrinsic motivation positively related to PA when external motivation was at a moderate level [69]. In addition, external motivation can make people feel regulated but if people agree with external viewpoints or values, external motivation can also promote their behavior [38]. Therefore, intervention for this subgroup should focus on amotivation and try to raise their awareness of PA, which makes them understand the positive value and significance of physical exercise.

Finally, the scores of intrinsic motivation and external motivation of Profile 3 (High Autonomous Motivation) were the highest while the score of amotivation was only lower than Profile 2. Furthermore, the intrinsic motivation score was higher than external motivation in Profile 3. This result is similar to previous research that intrinsic motivation and external motivation could positively affect students' PA [28]. However, some studies found that student's intrinsic motivation was positively associated with PA whereas external motivation could not predict PA among adolescents [46,70]. The reason for this difference may be that the influence of intrinsic motivation is more important than external motivation [71], and autonomy is crucial in PA among adolescents [72], so this subgroup has a positive attitude and intention toward PA.

## Practical suggestions

This study provides a novel sight for identifying latent categories of PA and offers some practical suggestions for improving PA among adolescents.

Firstly, appropriate physical activity has a positive effect on the healthy growth of children and adolescents. Therefore, the government or schools can set up training courses for parents to raise their awareness of the value of PA and increase their support for their children [73,74].

Secondly, the role of the school is also important in PA among adolescents. Early studies have shown that school PA support was significantly associated with student's participation in PA [75]. Therefore, principals and teachers should try to provide more opportunities and a better environment to increase student's engagement in sports [8,76].

Finally, intrinsic motivation is often linked with positive outcomes, including higher levels of enjoyment, satisfaction, and long-term commitment to sports [23]. Therefore, physical education teachers can appropriately reduce the difficulty of sports rules and increase games in physical education, thereby increasing their enjoyment and flow [77,78], which can promote students' sports engagement and decrease the drop out of students.

## Limitations and shortcomings

The current study is limited by some shortcomings. Firstly, although IPAQ-SF is considered an effective method to assess physical activity, the result measured by this tool may be higher or lower [79]. Therefore, future research can use some objective methods, such as an accelerometer, to measure students' physical activity. Secondly, this study only investigated adolescents in the Chinese Mainland and did not cover Western samples. Expanding studies should include students from other countries which can provide a more comprehensive view of the mechanism of adolescent participation in

PA. Finally, the result from LPA may be affected by external factors (such as society and environment) so further research should examine the stability of motivational profiles.

## Conclusion

In summary, the diversity of adolescent sports motivation types reflects individual differences and also suggests the need for more personalized strategies to promote adolescent participation in sports activities. The results suggested that the motivation of PA could be divided into three subgroups, and adolescents with higher intrinsic and external motivation were more active. In addition, future sports interventions need to pay attention to the phenomenon of low participation among students with low motivation, girls and older adolescents. This study offers a novel sight for adolescents' PA by latent profile analysis, which is valuable for informing intervention strategies for inactive adolescents.

## Author contributions

**Conceptualization:** Ying Zhao.

**Data curation:** Qinghua Wu.

**Formal analysis:** Ying Zhao.

**Funding acquisition:** Wei Zheng.

**Methodology:** Wei Zheng.

**Writing – original draft:** Ying Zhao.

**Writing – review & editing:** Wei Zheng.

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
