## [Decision Letter · Decision Letter 0]

PONE-D-25-04270

Motivation and Physical Activity across Chinese Adolescents:

Based on Latent Profile Analysis

PLOS ONE

Dear Dr. Zheng,

Thank you for submitting your manuscript to PLOS ONE. After careful consideration, we feel that it has merit but does not fully meet PLOS ONE’s publication criteria as it currently stands. Therefore, we invite you to submit a revised version of the manuscript that addresses the points raised during the review process.

We look forward to receiving your revised manuscript.

Kind regards,

Jindong Chang, Ph.D.

Academic Editor

PLOS ONE

Journal Requirements:

3. In this instance it seems there may be acceptable restrictions in place that prevent the public sharing of your minimal data. However, in line with our goal of ensuring long-term data availability to all interested researchers, PLOS’ Data Policy states that authors cannot be the sole named individuals responsible for ensuring data access (http://journals.plos.org/plosone/s/data-availability#loc-acceptable-data-sharing-methods).

Reviewers' comments:

Reviewer's Responses to Questions

**Comments to the Author**

1. Is the manuscript technically sound, and do the data support the conclusions?

Reviewer #1: Partly

2. Has the statistical analysis been performed appropriately and rigorously? 

Reviewer #1: Yes

3. Have the authors made all data underlying the findings in their manuscript fully available?

Reviewer #1: Yes

4. Is the manuscript presented in an intelligible fashion and written in standard English?

Reviewer #1: Yes

5. Review Comments to the Author

Reviewer #1: 1. The abstract need to be rewritten, specifically the Method section. It is too brief and did not explain the methodology of this study.

2. Check the in-text citation of the entire manuscript. Only last name and years need to be in your in-text citation.

3. The gap of the current study is unclear, and not spell out.

4. Current study chose the validated Chinese version of Sport Motivation Scale (Li & Harmer, 1996), however the newest/latest instrument in the field is Sport Motivation Scale-2. What is the rationale for ignoring the new version of SMS-2?

6. The smaller values of AIC, BIC, and aBIC indicate the better model fit… provide what the accepted small values.

7. “However, amotivation was not correlated with intrinsic motivation”… there are some irregularities in the explanation of this paragraph.

8. Conclusion: The word In ‘sum’ needs to be changed to In summary

9. Manuscript needs to undergo rigorous proofing reading.

6. PLOS authors have the option to publish the peer review history of their article (what does this mean? ). If published, this will include your full peer review and any attached files.

**Do you want your identity to be public for this peer review?** For information about this choice, including consent withdrawal, please see our Privacy Policy .

Reviewer #1: No

---

## [Author Response · Author response to Decision Letter 1]

13 Jun 2025

Dear Reviewer,

Thank you very much for your time and valuable feedback on our manuscript. We sincerely appreciate your constructive comments, which have helped us improve the quality and clarity of our work. Below, we provide a point-by-point response to your suggestions and outline the revisions we have made to the manuscript.

1. Abstract Rewriting

We have revised the Method section of the abstract to provide a clearer and more detailed explanation of the methodology. The updated version now includes key aspects such as the sample size, measurement tools (Sport Motivation Scale and IPAQ-SF), and analytical methods (LPA, MANOVA, ANOVA). This revision ensures that the abstract better reflects the study's design and procedures.

2. In-text Citation Format

We have carefully checked and corrected all in-text citations throughout the manuscript to ensure they adhere to the required format (last name and year only). Any deviations have been rectified.

3. Clarification of Research Gap

As suggested, we have expanded the discussion of the research gap in the Introduction section (page 7-8 of the revised manuscript). Specifically, we highlighted the lack of person-centered studies on motivation profiles among Chinese adolescents and the unique sociocultural context that may influence these profiles. This addition underscores the novelty and significance of our study.

4. Rationale for Using SMS Instead of SMS-2

We acknowledge the availability of the newer SMS-2 scale and appreciate the opportunity to clarify our choice. Our decision to use the original SMS was based on the following considerations:

Factor Structure: The SMS has demonstrated robust structural validity in prior studies, including those involving Chinese adolescents.

Internal Consistency: The scale’s reliability (Cronbach’s α) for all subscales in our study was acceptable (α > 0.70).

Dimensionality: The SMS-2 consolidates external motivation into a single dimension, whereas the SMS allows for a finer-grained analysis of introjection, identification, and external regulation-critical for our research questions.

6. Model Fit Indices (AIC, BIC, aBIC)

We have supplemented the explanation of model fit indices in the Results section (page 11). Specifically, we noted that differences >10 in AIC and >6 in BIC/aBIC are considered meaningful for model selection (Spurk et al., 2020; Tein et al., 2013). This addition clarifies the criteria used to determine the optimal profile model.

7. Description of Amotivation Correlations

We have revised the paragraph discussing amotivation correlations (page 13) to ensure accuracy and clarity. The updated text now clearly states that amotivation was not significantly correlated with intrinsic motivation to accomplishment or experience stimulation but showed weak positive correlations with identification and external regulation.

8. Conclusion Wording

The phrase “In sum” has been replaced with “In summary” in the Conclusion section.

9. Proofreading

The manuscript has undergone rigorous proofreading to correct grammatical errors, improve sentence structure, and ensure consistency in terminology. We have also double-checked tables and figures for accuracy.

Once again, we deeply appreciate your thoughtful review and hope that our revisions have addressed all your concerns satisfactorily. Please do not hesitate to contact us if further clarifications or adjustments are needed.

Best regards,

Wei Zheng

Sanming University

Email: zhengweiSanmingUniversity@outlook.com

---

## [Editor Report · Decision Letter 1]

Motivation and Physical Activity across Chinese Adolescents:

Based on Latent Profile Analysis

PONE-D-25-04270R1

Dear Dr. Zhang,

We’re pleased to inform you that your manuscript has been judged scientifically suitable for publication and will be formally accepted for publication once it meets all outstanding technical requirements.

Kind regards,

Jindong Chang, Ph.D.

Academic Editor

PLOS ONE
---

## [Editor Report · Acceptance letter]

PONE-D-25-04270R1

PLOS ONE

Dear Dr. Zheng,

I'm pleased to inform you that your manuscript has been deemed suitable for publication in PLOS ONE. Congratulations! Your manuscript is now being handed over to our production team.

Kind regards,

on behalf of

Dr. Jindong Chang

Academic Editor

PLOS ONE